# Drug Discovery Targeting Post-Translational Modifications in Response to DNA Damages Induced by Space Radiation

**DOI:** 10.3390/ijms24087656

**Published:** 2023-04-21

**Authors:** Dafei Xie, Qi Huang, Pingkun Zhou

**Affiliations:** 1Department of Radiation Biology, Beijing Key Laboratory for Radiobiology (BKLRB), Beijing Institute of Radiation Medicine, Taiping Road 27th, Haidian District, Beijing 100850, China; 2Department of Preventive Medicine, School of Public Health, University of South China, Changsheng West Road 28th, Zhengxiang District, Hengyang 421001, China

**Keywords:** radiation protection, space radiation, DSB, post-translational modification, drug target, drug discovery

## Abstract

DNA damage in astronauts induced by cosmic radiation poses a major barrier to human space exploration. Cellular responses and repair of the most lethal DNA double-strand breaks (DSBs) are crucial for genomic integrity and cell survival. Post-translational modifications (PTMs), including phosphorylation, ubiquitylation, and SUMOylation, are among the regulatory factors modulating a delicate balance and choice between predominant DSB repair pathways, such as non-homologous end joining (NHEJ) and homologous recombination (HR). In this review, we focused on the engagement of proteins in the DNA damage response (DDR) modulated by phosphorylation and ubiquitylation, including ATM, DNA-PKcs, CtIP, MDM2, and ubiquitin ligases. The involvement and function of acetylation, methylation, PARylation, and their essential proteins were also investigated, providing a repository of candidate targets for DDR regulators. However, there is a lack of radioprotectors in spite of their consideration in the discovery of radiosensitizers. We proposed new perspectives for the research and development of future agents against space radiation by the systematic integration and utilization of evolutionary strategies, including multi-omics analyses, rational computing methods, drug repositioning, and combinations of drugs and targets, which may facilitate the use of radioprotectors in practical applications in human space exploration to combat fatal radiation hazards.

## 1. Introduction

Deep space exploration and long-term human space missions are stalled and greatly restricted by hazards, including microgravity and cosmic radiation. Cosmic radiation is among the highest-priority threats to the health of astronauts [1,2,3]. Diverse ionizing radiations existing in the complex space environment, especially those of high linear energy transfer (LET), cause cataracts [4], promote Alzheimer’s disease [5], and affect cardiac physiology [6]. Crucially, increased cancer risks have been extensively reported [2,7,8]. Ionizing radiation exposure leads to chromosomal aberrations [9], DNA damage [10,11], alterations in the cell cycle [12,13], and apoptosis [14,15]. However, the complex nature and small astronaut cohort have made the research on space radiation protection challenging and the results highly uncertain [16]. The elusive mechanisms of ionizing radiation (IR)-induced DNA damage and repair have made research even more difficult. Shielding materials in space crafts are not sufficient, although they are still the main protective measures used. Medicine should play a more important role in protecting against space radiation, but no radioprotectors specifically counteracting the effects of space radiation, including high LET and chronic low-dose IR exposure, have been approved by the United States Food and Drug Administration (US FDA) [17,18]. The current protection technology is far behind the requirements of human space exploration. Thus, improvements in medical protection approaches against space radiation through the discovery and development of efficient agents with favorable toxicity profiles are urgently needed.

Elucidating the DNA damage response (DDR) facilitates research into radioprotectors. One of the ways in which space radiation damages DNA indirectly is through oxidative stress, with the production of reactive oxygen species (ROS) [19,20] and free radicals [21]. Antioxidants such as MitoQ decrease mitochondrial ROS [19], CBLB502 and trace elements scavenge free radicals [22,23], and vitamin A inhibits the expression of inflammation factors. They have been considered important radioprotective compounds [24]. Another kind of radioprotector is protease inhibitors, including ilomasta, which promotes the recovery of immunity [25], and Bowman–Birk inhibitors (BBI), which exert anticarcinogenic and anti-inflammatory properties [26,27]. Gamma-tocotrienol (GT3) and coenzyme Q10 (CoQ10) are also promising radioprotectors by preventing the apoptosis of cells [28,29,30]. Herbal mixtures, such as Hong Shan Capsule (HSC) [31] and resveratrol [32], were proven to be effective against IR. These agents lack structural diversity and have major drawbacks, including limited availability, uncertain safety profiles, and ambiguous mechanisms of action.

Direct IR-induced DNA damage is caused by the interaction of charged particles with DNA molecules [21], in which DNA double-strand breaks (DSBs) are extremely cytotoxic lesions [33,34]. DSBs can be repaired by several organized mechanisms to maintain the stability and integrity of the genome [35], which is vital for cell survival. Classical non-homologous end joining (NHEJ), homologous recombination (HR), alternative end joining (alt-EJ), and single-strand annealing (SSA) represent distinct DSB repair mechanisms [36], of which NHEJ and HR are the most pivotal and common. Post-translational modifications (PTMs) are covalent chemical modifications of proteins that occur after translation, conferring proper activity and biological functions to these proteins. The main PTMs related to DDR are phosphorylation, ubiquitylation, acetylation, methylation, SUMOylation, and poly ADP-ribosylation (PARylation). It was revealed that a number of DNA-damage-repair-related factors are subjected to these PTMs, which play indispensable roles in chromatin structures and functions. These factors lead to the rapid initiation and efficient regulation of a variety of biological processes by modulating DDR spatiotemporal dynamics [37,38,39,40]. Most PTMs are deposited on histones [41] and engage in the recruitment of a series of DDR proteins [38]. Targeting essential factors in the PTM of DNA DSB repair may lead to a promising strategy for developing radioprotectors for human space exploration, as the identification and verification of drug targets are the early and critical steps of drug discovery.

The aim of this review was to deepen our understanding of the detailed mechanisms of diverse PTMs in the repair of IR-induced DNA DSBs by integrating their essential participants and to take a global view of the involved proteins and complexes as potential drug targets against the effects of space radiation. We focused on recent findings related to molecular and cellular biological processes in the PTM of the DDR and summarized the crucial factors for each of them. We discussed their current utilization in radiosensitization and radioprotection and systematically analyzed their possibility as targets of potential radioprotectors in spaceflights. We emphasized the direction and paradigm in the research and development (R&D) of agents against space radiation exposure from a future perspective, highlighting the capability of large-scale computational and rational analyses.

## 2. Literature Search and Curation

The widely used search engine PubMed (https://pubmed.ncbi.nlm.nih.gov/, accessed on 7 September 2022) was employed for a literature search and collection with the keywords of “DNA double-strand break”, “post-translational modification”, and “space radiation”. It was completed in September 2022 and supplemented in April 2023, and limited to English-language literature. Original research articles and reviews were considered. We investigated the collected literature and extended our list of references by examining their bibliographies. As a result, a total of 241 papers were included in this review, among which 82.2% (198/241) were published after 2010. Our main intention is (i) to communicate the research areas of mechanisms of PTMs in DSB repair and medical protection against space radiation, and (ii) to help us with future directions and clues in the R&D of potential radioprotectors to be used in the space environment.

## 3. PTMs in the Choice of DNA Repair Pathways

HR is a critical pathway for the error-free repair of DNA DSBs, while NHEJ always occurs in the absence of a sister chromatid, leading to error-prone repair and more mutations [42,43,44]. NHEJ was reported to be the predominant DNA repair pathway in mammalian cells [45]. The choice of the repair pathway was found to be tightly associated with the cell cycle, as NHEJ is the default repair pathway [46] usually executed in the G1 phase of the cell cycle in a rapid and high-capacity manner [42,47]. Unlike NHEJ, which may occur throughout the entire cell cycle, HR is largely limited to the S/G2 phases [42,48] and is conducted more slowly than NHEJ [47]. The underlying mechanism is that DSB repair is executed with higher efficiency during the S phase. DSB processing and checkpoint activation are much more efficient in the G2/M phase than in the G1 phase [49]. In general, the 5′-3′ degradation of DSB ends is needed for the loading of checkpoint and recombination proteins in all HR reactions [50,51]. The generation of long 3′ single-strand DNA (ssDNA) overhangs mediated by DNA helicases and exonuclease in DNA end resection was proven to be an essential committed process in HR [47,48,52,53,54]. In contrast, NHEJ requires DNA ends that have not been resected instead of 3′ ssDNA tails. DNA end resection is not needed, leading to the joining of two DNA ends with few references to the DNA sequence [47,48,55,56]. Therefore, controlling DNA end resection is one of the processes affecting whether DNA repair is conducted by NHEJ or HR [35,40]. For example, the 53BP1-RIF1-shieldin complex cooperates with the CTC1-STN1-TEN1 (CST)/Pol α-Prim complex in regulating the generation of 3′ overhangs, which are essential for DNA end protection and switching the DSB repair mode to NHEJ [53,57]. In contrast, BRCA1 promotes HR and antagonizes NHEJ by stimulating end resection [58,59]. Several key proteins and their complexes play regulatory roles in NHEJ. For instance, the Ku70-Ku80 heterodimer is central in initiating NHEJ by recognizing DSB ends and recruiting DNA-PKcs to DSB sites [51,60,61,62,63]. In addition, 53BP1 stimulates NHEJ by recruiting other DDR proteins such as ATM and inhibiting DNA end resection processing by protecting broken DNA ends with its co-factors PTIP, RIF1-shieldin, or REV7/MAD2L2 [48,56,61,62,64,65,66,67]. In contrast, the important factors in HR mainly include BRCA1/2, EXO1, MRE11 [47,48,64,68,69], and RAD51 and its paralogs [43,47,69,70]. Among them, BRCA1 directly affects the DSB repair pathway choice by regulating the initiation of end resection [52,59]. The preservation of long-term resection activity requires EXO1 exonuclease [71], the deficiency of which contributes to the accumulation of unprocessed DSBs and HR failure [72]. MRE11 exonuclease activity is needed for the assembly of a series of proteins to DSB sites to mediate extended-end resection for HR [73].

HR is orchestrated by several PTMs with elaborate primary mechanisms. The first one is phosphorylation. Switching the meiotic recombination mode of HR was reported to occur by the phosphorylation of RAD54 and HED1, downregulating RAD51 activity by suppressing Rad51/Rad54 complex formation [74,75]. Secondly, SUMOylation is important in HR. It affects all steps in HR and exerts various regulatory functions on substrates [76]. Evidence indicated that SUMOylation induced by topoisomerase 1-binding arginine/serine-rich protein (TOPORS) was essential for the recruitment of RAD51 to the damaged sites and the support of HR repair, maintaining genomic stability [77]. On the other hand, NHEJ might be associated with phosphorylation and methylation by DNA-PKcs and 53BP1, respectively.

NHEJ and HR are competitive, and their balance is finely modulated by bioprocesses that include PTMs. Ubiquitination is the most vital PTM, playing a specific role in the recruitment and enrichment of DDR factors at DSB sites in chromosomes and governing DNA repair pathway choices between NHEJ and HR. DDR proteins are mainly assembled by ubiquitin E3 ligases RNF8 and RNF168, followed by accurate repair processes [35]. The ubiquitylation-dependent DSB repair pathway choice is frequently associated with DNA end resection. For example, Cullin3-KLHL15 ubiquitin ligase participates in CtIP protein turnover through the ubiquitin–proteasome pathway, fine-tuning DNA end resection and impacting the balance between HR and NHEJ [78]. RING domain-containing E3 ligase RNF138 is involved in the ubiquitination of Ku80 during the S phase and its removal from DSB sites, stimulating DSB end resection and promoting HR initiation [79]. In addition to DNA end resection, ubiquitylation also modulates the choice of DNA repair pathways by altering the expression of specific DDR proteins. CtIP, which is a target of anaphase-promoting complex/cyclosome (APC/C) ubiquitin ligase, is downregulated during G1 and G2 phases and reduces HR [80]. CtIP ubiquitylation and upregulation are stimulated by UBE2Ds and RNF138 at DNA damage sites, promoting DNA repair by HR [81]. Moreover, ubiquitination affects the DNA repair pathway choice by regulating histone H2A at Lys15 (H2AK15ub) and initiating downstream signaling events [82]. Phosphorylation is another major PTM involved in the balance of DNA repair pathways. The phosphorylation of ubiquitin at Thr12 (pUbT12) influences DDR by regulating the activity of 53BP1 in damaged chromosomes [83], and 53BP1 inhibits excessive DNA end resection and promotes repair by NHEJ through different phosphoprotein interactions [84]. RIF1 is prominent at DSB sites in the G1 phase of the cell cycle by the ATM-associated phosphorylation of 53BP1, ensuring the dominant position of NHEJ in this phase [58,59,85].

Collectively, a variety of proteins and their complexes were revealed to act in the complicated response mechanisms to DNA lesions induced by IR, participating in distinct PTMs and coordinating NHEJ, HR, and their balance in DNA repair (Figure 1). These essential factors could be properly categorized and investigated from the view of PTMs, including phosphorylation, ubiquitylation, acetylation, and methylation. Compounds targeting these factors influence DNA repair after IR, leading to radiosensitization or radioprotective effects (Table 1). Some of them have been approved for regulating DDR, and more candidates are under development. They provide resources in the discovery of future space radioprotectors (Figure 2).

## 4. Targeting Essential Phosphorylation Factors for Regulating DDR

The essential participants in phosphorylation, which regulates the alterations of protein conformation and biological activity, include ATM, DNA-PKcs, CtIP, ATR, CHK1, H2AX, and RAD51/52/54.

ATM is one of the major and most extensively studied kinases involved in the early stages of cellular responses to DNA DSBs [86]. ATM is activated in the phosphorylation of H2AX [86,87]. Many ATM-dependent phosphorylation processes need sustained activity of ATM, and a phosphorylation site on ATM itself functions in its retention on damaged chromatin [159]. ATM mediates the rapid phosphorylation of DNA-PKcs at Thr-2609 and the adjacent (S/T) Q motifs within the Thr-2609 cluster upon IR-induced DNA DSBs [88], as well as MDM2 phosphorylation at multiple sites near its RING domain [89]. In contrast, mutation at lysine 3016 of ATM inhibits the phosphorylation of p53 and CHK2 [90], which was reported to play a key role in stabilizing p53 [89]. ATM has been considered a drug target because increased ATM is related to the development of radioresistance [160]. Thus, this kinase has been frequently investigated in attempts to improve radiation therapy (RT) in cancer treatment [161,162]. It is inhibited by radiosensitizers [163] such as 2-hydroxy glutarate [91] and AZD1390 [92]. It is also inhibited by caffeine, according to DrugBank [164]. A considerable number of ATM inhibitors are under development as components of future cancer therapies [165]. A radioprotector named isorhamnetin, targeting ATM, was reported in 2021. It prevents IR-induced gastrointestinal syndrome by promoting ATM phosphorylation and enhancing the recruitment of 53BP1 [93].

An important partner of ATM is DNA-PKcs. DNA-PKcs phosphorylation mediated by ATM is needed for the full activation of DNA-PKcs and subsequent DSB repair [88]. DNA-PKcs targets phosphorylation sites on ATM, and phosphorylation mutations markedly inhibit ATM activity and affect the ATM signaling of DSBs [166]. DNA-PKcs is highly relevant to cancer. Previous studies reported its involvement in cancer metabolism by impacting glycolysis [167], emphasizing it as a promising therapeutic target. Interestingly, different regulatory co-factors induce distinct DNA-PKcs phosphorylation kinetics at Thr-2609 and Ser-2056, playing primary roles in DSB repair and the establishment of cellular radioresistance [88]. Thus, DNA-PKcs is often targeted for overcoming radioresistance in cancer therapies. For example, the triple-target (DNA-PK/PI3K/mTOR) inhibitor PI-103 was found to increase the radiosensitivity of a glioblastoma cell line subtype by targeting DNA-PKcs [94]. The sensitivity of several other kinds of cancer cells to RT and chemotherapy could be significantly promoted by NU7441 as a DNA-PKcs inhibitor [95,96]. According to DrugBank, DNA-PKcs is targeted by caffeine and SF1126. The latter is an investigational agent at present for the treatment of various forms of cancers. Decreases or deficiencies in DNA-PKcs lead to accelerated cellular senescence [168], which offers clues for protecting cells from radiation-induced premature senility.

One of the sophisticated molecular switches controlling the balance between NHEJ and HR in DSB repair is the phosphorylation of serine 327 in CtIP throughout the cell cycle [42]. DNA end resection was found to occur not only during HR but also in NHEJ in the G1 phase in a distinct manner controlled by CtIP [97]. In this process, CtIP is phosphorylated by PIK3 and interacts with BRCA1, stimulating the initiation of resection [169]. CtIP is among the proteins participating in the resections of the late and persistent DSBs. CtIP is made use of in cancer treatment, since it increases p38-MAPK reactivation in cooperation with CHK1 in response to RT [170]. The ATP-competitive mTOR inhibitor torin2 reduces the number of radiation-induced CtIP and RAD51 foci formed, indicating that this compound radiosensitizes cancerous tumors by blocking HR. Mechanically, it causes an S-phase-specific DNA repair deficiency [99]. Regarding radioprotection, CtIP was reported to function in DSB repair by NHEJ in the G0/G1 phase through phosphorylation at Thr-847 [171]. The engagement in DNA repair implied the potential use of CtIP in preventing and relieving radiation injuries.

ATR and CHK1 are common participants in phosphorylation. ATR phosphorylates several BRCA1 fragments directly in response to IR [100]. Interactions between these two proteins have also been observed. The ATR-CHK1 pathway was implicated in DDR processes [172]. CHK1 phosphorylation at serine 345 is suppressed by the overexpression of ATR in mutated cells and enhanced in wild-type cells [173]. ATM and the nuclease activity of MRE11 are required for ATR recruitment, followed by the phosphorylation and activation of CHK1 [174]. These two proteins have been identified as ideal therapeutic targets in cancer treatment, and their inhibitors have been in clinical trials either as single drugs or in combination with other genotoxic agents [175]. A growing number of research studies proved the efficiency of ATR inhibitors as viable anticancer drugs. Eight of them are currently under development [101], among which ceralasertib is included in DrugBank. These kinds of inhibitors were shown to be effective, especially in the treatment of head and neck squamous cell carcinoma, in combination with other therapies such as surgery, RT, and chemotherapy [176]. They also specifically target DDR pathways and impair DSB repair processes, exhibiting significant radiosensitizing effects [177]. CHK1 is also related to as many as thirty-eight compounds, according to DrugBank, the majority of which are investigational or experimental. The only approved drug is fostamatinib, which is used in the treatment of chronic immune thrombocytopenia. Nevertheless, targeting CHK1 represents an attractive strategy for potentiating the efficacy of RT. DDR proteins, including CHK1 and ATR that promote HR, are upregulated in radioresistant breast cancer cells, and the CHK1 inhibitor AZD7762 sensitizes these cells to IR [98]. Another CHK1 inhibitor, PF-477736, was proven to enhance the radiosensitivity of human triple-negative breast cancer cells [102]. Nexrutine was reported to increase the sensitivity of prostate cancer cells to IR by reducing the expression of several proteins, including CHK1 [103]. However, no radioprotectors have yet been developed based on ATR or CHK1.

RAD51 and its paralogs function in the transduction of DNA damage signaling and facilitate the repair of DNA breaks [43]. RAD51 is believed to be among the central factors in HR repair and regulated at the level of PTM in this pathway. The most significant PTM related to RAD51 is phosphorylation, which mediates distinct functions in promoting HR in response to global DNA damage [37]. Phosphoaminophosphonic acid-adenylate este and amuvatinib were collected in DrugBank targeting RAD51. The latter is in trials for treatment of solid tumors. Evidence shows that targeting RAD51 contributes to overcome the radioresistance of some kinds of cancers [70,104,105], such as methotrexate inhibiting RAD51 expression and radiosensitizing human osteosarcoma cells [106]. A RAD51 paralog, RAD52, was implicated in phosphorylation at T412, facilitating later stages of HR [37,178]. Its enhancement elicits the radioresistance of cancer stem cells [107] and it was widely explored in synthetic-lethality-based anticancer therapies [179]. RAD54, which participates in polymerase-dependent DNA synthesis and the completion of HR [180,181], was found to be phosphorylated by NEK1 during the G2 phase and by CDK2 to limit its branch migration activity in HR [37,182]. However, reports on its use in radiosensitivity regulation are insufficient.

Histone H2AX is one of the regulators of the checkpoint pathways responding to DNA DSBs [108]. It is mainly phosphorylated by ATM at serine 139 in the early stages of DNA DSBs and forms foci at the sites of DNA damage [86,183]. H2AX phosphorylation contributes to DDR, and the ability to enhance H2AX phosphorylation decreases substantially in DNA-repair-deficient cells after IR [183]. H2AX also facilitates 53BP1 recruitment to DNA break sites [109]. However, there are few reports on the application of H2AX in regulating radiation-induced DDR.

Other crucial factors in phosphorylation include BRCA1, CHK2, and the MRE11-RAD50-NBS1 complex (MRN), as well as the downstream effectors p53, NF-κB, AKT, and survivin [184]. In addition, bioprocesses such as autophosphorylation [47,185], hyperphosphorylation [100,186], and dephosphorylation [159] all play important roles in the regulation of the multi-protein network, which is irreplaceable for the maintenance of genomic integrity [187]. An extensive overview and deep understanding of them may contribute to an elucidation of the detailed molecular mechanisms of cellular DDR to different kinds of radiation and the discovery of novel regulators by targeting signal pathways related to phosphorylation.

## 5. Ubiquitylation and Its Roles in Regulating DNA Repair

Ubiquitylation is a kind of reversible covalent modification that modulates the activity of many DDR proteins [140] and participates in the regulation of the multi-protein network to ensure genomic stability [187]. The ubiquitin–proteasome pathway is needed for forming replication-dependent DSBs in cells treated with chemotherapeutic drugs [188]. Moreover, histone ubiquitination indirectly regulates the recruitment of many DDR proteins to damage sites in DSB repair processes [109]. The typical factors in ubiquitylation are RNF8, RNF168, REV1, MDM2, and the BRCA1/BARD1 complex.

RNF8 and RNF168 are RING-finger E3 ubiquitin ligases that initiate DDR-related signal pathways [110] and are involved in the generation of binding sites for DNA repair factors [111]. RNF8 mainly mediates UBC13-dependent ubiquitylation at DNA damage sites [111]. It rapidly assembles at DSB sites in response to DNA damage by interacting with the adaptor protein MDC1 and activates H2A/H2AX ubiquitination with K63-linked ubiquitin chains. This bioprocess ensures genomic integrity by allowing the accumulation of DSB repair proteins in DSB-flanking chromatin near DNA lesions [110,189]. RNF168 specifically catalyzes histone ubiquitination on K13-15 [47,110,111,113], and RNF168 deficiency causes dysregulation in replication fork progression and restarting [113]. RNF168 interacts with ubiquitylated histone and is involved in the assembly of 53BP1 and BRCA1 at DNA break sites [114,115]. Furthermore, 53BP1 is recruited downstream of RNF168 as a reader of the H2A ubiquitin marker [65], while BRCA1 ubiquitylates claspin, an essential coactivator of CHK1 [190]. Canonical DSB ubiquitination is suppressed by ATM by blocking the accumulation of RNF168 and BRCA1 [191]. RNF8 and RNF168 are under development for utilization in antitumor therapies. A high expression of RNF8 was found in triple-negative breast cancer and was associated with a poor prognosis, demonstrating its potential as an anticancer drug target [112]. The 1,2,3-triazole derivative of quinazoline 5a, which may serve as a novel radiosensitizer, directly binds to RNF168, leading to reduced H2A ubiquitination mediated by RNF168 and impaired HR [116]. None have been exploited for radioprotection.

REV1 is a member of the Y-type DNA polymerase family in eukaryotes engaged in DNA damage tolerance, which plays a non-catalytic role independent of RAD18 in the modulation of translesion synthesis, leading to the development of radioresistance [117,118,119]. REV1 interacts with ubiquitin via ubiquitin-binding motifs at its C-terminus, which improves the relationship between ubiquitylated PCNA and REV1. The function of ubiquitin-binding motifs determines the association of REV1 with replication foci and impacts DNA damage tolerance and DSB-induced mutagenesis [120]. As a regulator of DNA damage tolerance, the sensitizing effect of REV1 in cancer therapy was investigated, especially its roles in radioresistance. However, research on the possible further applications of REV1 in radioprotection, except for its protective effect against ultraviolet light, is rare [192,193].

MDM2 coordinates with p53 and ATM after DNA damage via ubiquitylation, and p53 polyubiquitination is selectively inhibited by blocking MDM2 oligomerization, which provides a scaffold for the extension of polyubiquitin chains. The capacity of MDM2 to polyubiquitinate p53 is suppressed by ATM near the RING domain, sustaining its stability. The action of MDM2 self-ubiquitination and MDMX ubiquitination lead to the retention of E3 ligase activity [89]. Six drugs target MDM2, according to DrugBank, four of which have been approved by the US FDA. They are all diverse forms of zinc. MDM2 was shown to be highly expressed in brain and lung cancer, and its genetic inhibition suppressed the growth of tumor cells. Enhancing the radiosensitivity of different kinds of tumors by targeting MDM2 is also considered a rational therapeutic strategy [121,122,123]. The combined use of MDM2 and Bcl-2/XL/W inhibitors was reported to have synergistic effects in anticancer treatments [124]. However, MDM2 has not been explored for its potential use in radioprotection.

Another crucial factor in ubiquitylation is the BRCA1/BARD1 complex, which elevates the recombinase activity of RAD51 and stimulates the late process of HR [83,125]. Interactions between this complex and its agonists and antagonists are implicated in the formation of cancers related to deficient or abnormal chromosomal damage repair and the maintenance of replication forks. This functional complex has been explored for its therapeutic use in tumor suppression [126]. Notably, metformin, active against type 2 diabetes, was proposed as an effective candidate for alleviating senescence and cardiotoxicity induced by IR. It enhanced the expression of the BRCA1-BARD1-RAD51 complex and promoted BARD1-related DSB repair [127]. Thus, it might be repositioned for antiradiation use.

RAD51 is noteworthy, since RFWD3-dependent ubiquitination on it determines the radio-bidirectional effect. An anticonvulsant valproate decreases HR-associated RAD51 and enhances RFWD3 for the radiosensitization of cancer cells. Its functions in normal cells are opposite for radioprotection [128], which suggests its potential use in aeroastromedicine. RAD54 is also a prominent protein subjected to ubiquitylation-dependent destruction induced by APC/C in the G1 phase, which is supposed to be particularly associated with meiotic HR [37,194].

Monoubiquitylation and polyubiquitylation in DDR are also noticeable in spite of the fewer studies. DNA damage signals induced by genotoxic agents such as radiation and chemotherapeutic drugs need ubiquitin, ubiquitin-activating enzymes (E1), ubiquitin ligases (E3), and the formation of polyubiquitin chains on topoisomerase 1 (TOP1) [188]. E3 activity of TRAF6 or the X-linked inhibitor of the apoptosis protein is promoted by ATM exported from the nucleus after IR. It mediates the auto-polyubiquitylation of TRAF6 and the polyubiquitylation of ELKS [195]. These bioprocesses are complex with many steps and need further exploration. There is no doubt that targeting ubiquitylation will provide more varied options for modulating cellular responses to IR.

## 6. DNA Damage Repair Modulated by Other PTMs

Acetylation plays an indispensable role in the comprehensive map of the chromatin landscape. A DNA-damage-induced acetylation-related switch is supposed to be appropriately regulated by the SAGA complex at histone H2B lysine 120 [196]. Tip60 is a histone acetyltransferase involved in activation of ATM at lysine 3016 as a substrate in acetylation located in the highly conserved C-terminal FATC domain. This bioprocess is a key step in DSBs and ATM activity. The acetyltransferase activity of Tip60 is activated by its direct interplay with histone H3 trimethylated on lysine 9 (H3K9me3) at DSB sites [90,129]. Tip60 is targeted by coenzyme A and S-Acetyl-Cysteine in DrugBank, which were not designed for the regulation of cellular responses to radiation and have not been approved yet. Targeting mutations of Tip60 may sensitize cancer cells to IR and PARP inhibitors (PARPi) [197]. The Tip60 inhibitor NU9056 was proposed as a potent candidate against prostate cancer, as it reduced Tip60 stabilization after IR [130]. This acetyltransferase could also play a role in the discovery of radioprotectors. For instance, the GSK-3 inhibitor CHIR99021 protects intestinal stem cells from IR in vivo by specifically preventing Tip60-induced apoptosis [131].

Aberrant histone methylation may lead to tumorigenesis and its progression by affecting DSB repair efficiency [129]. On the other side, methylation is also induced by IR and DNA damaging chemicals that have been widely used in the treatment of cancers. Adducts of methylation trigger DSBs and finally apoptosis of cells. The lesions often occur in the narrow area between methylated nucleosomes [184,198]. Several symbolic factors in methylation exert influences on DDR. They may be utilized in cancer prognosis and treatment. For example, BRCA1 promoter methylation indicates PARPi resistance of triple-negative breast cancer cells, which makes germline BRCA1 mutations the most reliable biomarkers in predicting tumor response to the PARPi-based therapies [126]. Furthermore, we found that BRCA1 was intensively explored for its regulatory role and potential use in radiosensitization in RT in recent years [132,133]. As for radioprotection, tyrosine kinase inhibitors dasatinib and imatinib were found to be radioprotective by increasing DSB repair. BRCA1 was upregulated as an HR-related gene in this bioprocess [134]. Dimethyl sulfoxide was also found to attenuate IR-induced injury to the male reproductive system. One of the main mechanisms was that this compound elevated the expression of phosph-BRCA1, BRCA1, and RAD51, all of which indicated enhanced DSB repair with a bias toward HR [135]. In addition to BRCA1, 53BP1 is another typical protein related to methylation. Recognition of histone H4 Lys 20 (H4K20) methylation by the tandem tudor domain of 53BP1 is indispensable for its efficient recruitment to DSB sites [65,109]. It was reported that the 53BP1-related pathway could be activated by voriconazole in human keratinocytes. It potentiated ultraviolet-light-induced DNA damage [137], suggesting its potential use in radiosensitization. The pattern by which 53BP1 impacts DDR is supposed to be associated with oxidative stress and ROS. The levels of 53BP1 and ROS could be significantly decreased by inorganic nitrate after IR in mice, which relieved IR-induced systemic damage markedly and exerted a radioprotective effect [136].

PARylation is a primary event catalyzed by PARPs in response to DSBs in a few seconds after IR. It occurs at the adjacent regions of DSBs and facilitates the accumulation of DDR proteins to the break sites via their poly ADP-ribose (PAR)-binding domains [138]. PARP1 is recruited to the DSB sites and PARylation eventually mediates release of it. This polymerase is of great importance in the feedback regulation of PARylation, determining the rapid assembly of DNA repair factors to maintain genome stability [199]. As many as twenty-two compounds target PARP1 according to DrugBank, among which its inhibitors olaparib, rucaparib, niraparib, and talazoparib have been approved or are under development for treatment of various cancers [139].

Neddylation, a significant protein function regulatory biochemical process, has been proved to stimulate ubiquitylation of Ku and its release from DNA break sites after DSB repair. Neddylation was illustrated to be enhanced in multiple human cancers, implying its potential use in the discovery of novel anticancer drugs [142,200]. The ubiquitin-like protein NEDD8 is added to substrate proteins in this PTM [201]. Its recruitment and conjugation are highly dynamic processes that could be modulated by a few small molecules, such as MLN4924, an E1 NEDD8-activating enzyme inhibitor [140,141]. This compound is in clinical trials against several kinds of tumors by the induction of cell cycle arrest, apoptosis, senescence, and autophagy [141,143]. In addition, gossypol, an inhibitor of cullin neddylation, also exerts a synergistical anticancer effect with the specific MCL1 inhibitor [142].

SUMOylation and deSUMOylation are important PTMs in DDR, being focused on partly for their modulation of RT resistance [202]. For example, the use of SUMOylation inhibitors 2-D08 and ML-792 significantly enhances the sensitivity of cancer cells to RT and chemotherapy [144]. SUMOylation of MEIIL3 was reported to induce the overexpression of specific miRNA and proteins in colorectal cancer, suggesting its possible therapeutic role in genetic mutation and tumorigenesis [145]. In addition, a small-molecule radiosensitizer YTR107 was found to suppress SUMOylated NPM1 from interacting with RAD51, resulting in impaired DSB repair and reduced radioresistance of cancer cells [146]. Furthermore, RAD52 was proved to be SUMOylated when DNA damage was sensed, preserving genomic stability against proteasomal degradation [203,204]. However, SUMOylation has rarely been investigated for its potential use in radioprotection.

Glycosylation is another PTM in the cellular response to DSBs, and O-linked β-N-acetylglucosamine glycosylation is of special importance in HR and SSA. Currently, O-GlcNAc transferase (OGT) is the only known enzyme in this process affecting cell survival by modulating cell cycle and RAD52 functions. It has been proposed as a novel target in cancer therapies [147,148]. In addition, NEIL3 DNA glycosylase was extensively studied. Cleavage by this base excision repair enzyme was proved to be the primary unhooking mechanism in DNA interstrand cross-link (ICL) repair [149,150]. NEIL3 mediates progression and radio/chemosensitivity of prostate cancer and thus was considered as a potential therapeutic target [151,152]. The RUVBL1/2 complex and TRAIP turned out to be indispensable for the NEIL3 pathway [150]. DrugBank indicates RUVBL2 as a target of quercetin. In fact, glycosylation was investigated for its roles in radioprotection, and glucosylation showed positive effects. Antioxidant glucosyl flavonoids including quercetin, naringenin, and hesperetin displayed reduced DNA DSB formation in vitro, suggesting their radioprotective properties [153].

Lysine crotonylation (Kcr) is an emerging PTM correlated widely to transcriptional silencing, DSB repair, and DNA replication stress response, the level of which is altered by DNA damage and replication stress [205]. For example, Kcr levels of a series of DSB repair proteins such as RPA1, POLD1, and XRCC5 are upregulated prominently after DNA damage. Among them, the Kcr of RPA1 promotes HR, while it occurs independently with transcriptional silencing [206,207]. In transcriptional repression, CDYL1 edits the histone code [154]. The major histone crotonyltransferases include CBP/P300, MOF, and GCN5 [155,156,208]. According to DrugBank, both P300 and GCN5 are targeted by the investigational coenzyme A. Interference of CBP/P300 may contribute to overcoming the therapeutic resistance of radiotherapy, representing a promising target for anticancer treatment [157,158]. Deeper exploration into Kcr in DDR, especially targeting it for radioprotective use, is rare up to now.

Other canonical PTMs include proteolysis, S-nitrosylation, and lipidation. Proteolysis-targeting chimera (PROTAC) is an engineered technology for targeted protein degradation, which has been applied in drug discovery, particularly for anticarcinogens [209,210]. USP37 regulates DDR, the deficiency of which induces aggravated DNA damage by accelerated proteolysis of BLM [211], indicating possible radiosensitizing uses. BLM is targeted by four compounds in DrugBank, two of which have been approved: antibacterial amoxicillin and citric acid as a kind of anti-chelation flavoring and preservative agent. S-nitrosylation is an oxidative modification of cysteine by nitric oxide (NO) and involved in cancer progression. Focusing on S-nitrosylation might be beneficial in improving cancer therapies [212,213,214]. Lipidation affects the conformation, stability, localization, and binding activity of proteins, which was found to be associated with various diseases, including neurological disorders, metabolic diseases, and cancers [215]. However, more work is still needed to explore their participation in DDR, especially radioprotection.

## 7. Conclusions and Future Directions

Exposure to cosmic radiation, which induces severe DNA damage, poses one of the major obstacles to human space exploration [216,217]. DSBs are considered to be highly toxic lesions causing genetic instability that should be properly repaired for cell survival [34,218]. NHEJ and HR are two predominant mechanisms for DSB repair in eukaryotic cells [219]. We found that their completion relies on several PTMs, including phosphorylation, SUMOylation, and methylation. A delicate shift between these two pathways involves sophistical modulation of ubiquitylation and phosphorylation. Functional proteins and complexes participating in DDR are intensively involved in PTMs. The essential participants of phosphorylation include ATM, DNA-PKcs, CtIP, and ATR. Ubiquitylation is mainly related to a series of ubiquitin ligases and polymerases. Acetylation, methylation, PARylation, neddylation, SUMOylation, glycosylation, and Kcr are all proven to systematically coordinate complicated biochemical processes in response to DSBs induced by IR. These PTMs and their associated proteins have provided a library and repository for our efforts to regulate and alter cellular responses to IR, including space radiation. Currently, they are being explored for use in cancer therapies, especially for improvements in radio- and chemosensitization. Their engagement in radioprotection has rarely been discussed, although their use in relieving the side effects of RT has sporadically been reported. None of them have been especially investigated for radioprotection in space exploration, leading to an extreme lack of available agents protecting astronauts from the complicated spatial radiation environment (Table 2). Of note, Kcr has attracted great attention and interest from academia. Thus, future illumination of the detailed mechanisms underlying DDR and the promotion of radioprotection technology based on Kcr are expected.

In the future, the rapid discovery and development of protective drugs against space radiation might rely more on the progression of multi-omics, including transcriptomics, epigenomics, metabolomics, and signalosomes. This technology will provide a comprehensive landscape of integrated knowledge from a variety of perspectives and new insight into understanding the underlying molecular mechanisms of the physiological and pathological responses to IR-induced DSBs. Advances in methodologies, such as high-throughput and high-content screening, will accelerate the accumulation of material and data on multiple levels for further analyses. The systematic integration and interpretation of data will provide deeper insight into the complete molecular functions and biochemical processes in response to IR. Taken together, making full use of multi-omics methods may bring a wave of new changes, including innovation in the promotion of technology in radioprotection by mapping information from multiple disciplines and cross-disciplines. This strategy is revolutionary and might expand the frontier of our recognition of DDR. Novel and potent agents could be proposed and evaluated as future radiosensitizers and radioprotectors based on this strategy.

The identification of drug targets is widely believed to be one of the most significant steps in the systematic research of novel drugs. The roles of computational biology and systems biology in predicting drug targets and the screening of candidate drugs have been highlighted in the post-genome era. Rational analysis and deliberate design based on big data and large-scale computation will contribute to more successes by reducing blindness and enhancing efficacy in drug tests and trials. It is the right time to introduce computing methods to the discovery of radioprotectors, as public resources such as databases and literature are abundant. For instance, the well-established Gene Expression Omnibus (GEO) is a functional data repository of high-throughput gene expression data from genomic hybridization experiments [220]. The GEO contains the expression profiles of DDR proteins after different kinds of radiation in several human cell lines. The Human Protein Reference Database (HPRD) provides detailed information on functional proteins in various biological processes and contains 41,327 protein–protein interactions and 93,710 PTMs [221]. The Reactome knowledgebase integrates data on signal transduction, DNA replication, and metabolism, establishing their connections and constructing an ordered network of molecular transformations [222]. The famous Gene Ontology (GO) knowledgebase is a dynamic ontology collecting information on gene functions with respect to three independent aspects, namely biological processes, molecular functions, and cellular components [223]. The Kyoto Encyclopedia of Genes and Genomes (KEGG) is a reference database for the biological explanation of high-level functions of cellular organisms and provides widely used pathway maps [224]. The well-curated pharmaceutical database DrugBank contains the most comprehensive information on 15,234 drugs and their targets [164]. Another large-scale pharmacogenomics dataset of small molecules is the Library of Integrated Network-based Cellular Signatures (LINCS), which catalogs cellular signatures in response to a variety of chemical perturbations. It also includes featured tools for further analysis and visualization [225]. More diverse and powerful algorithms and software could be exploited and employed in the construction and interpretation of networks or motifs, text mining with intelligent computational methods, and predictions and analyses conducted by machine learning.

One of the difficulties in the pharmaceutical industry is that the prevailing paradigm of drug discovery is extremely time-consuming, expensive, and risky [226]. Drug repositioning, which is conducted to identify different therapeutic indications for known compounds, has shown excellent safety profiles [227] and promising effects. Accordingly, more drug discovery initiatives will incorporate this attractive strategy, since it reduces time, costs, and risks of early drug research [228] and expedites the approval of new indications, including radioprotection, for old drugs. We believe that this strategy will also be extraordinarily useful for the fast development of radioprotective agents in human space exploration. Currently, well-acknowledged methods for drug repositioning are biological experimental approaches, computational approaches, and their combination [229]. Different kinds of computational approaches exist. They are based on the knowledge of targets, drugs, and diseases by signature and pathway/network-centric methods [230]. Potential repositioned drug targets can be discovered via the pipelines of molecular docking and similarity comparisons of drug structures. Machine learning has also been applied to construct predictive models and frameworks using the structures and molecular properties of drugs [231,232,233]. A workflow based on network analyses has been established and can predict possible drug–target interactions, providing clues for drug repositioning [234,235,236]. In recent years, several genes were proposed by integrative coexpression network analyses as potential targets of repositioned drugs against acute radiation syndrome according to the target-based approach [237]. Drug–drug relationships can be investigated by comparing side effect similarities [238] and transcriptional responses. Gene expression signatures, in addition to literature-based discovery, could also be used in the disease-centric approach. Successful radioprotection based on drug repositioning includes the non-steroidal anti-inflammatory sodium diclofenac [239], the GABA-ergic agonist baclofen [240], and the local anesthetic prilocaine hydrochloride [241].

The far-seeing plan on the strategy against space radiation should contain more concentration on the potency, availability, and oral bioavailability of drugs and their interactions, such as synergistic effects. Combinations of radioprotectors or multi-target drugs provide greater prospects. They will play more important roles in radioprotection, as they could improve the antiradiation efficacy at lower doses. It should be noted that the results of combined therapies are inferred to be complicated and unpredictable, which suggests careful consideration and balancing. Moreover, possible drug–drug interactions and the resulting adverse effects should also be looked into. The future treatment regimens with appropriate collocation of multi-target drugs might provide the best possible combinations.

In conclusion, the pharmaceutical industry suffers from some current predicaments, including the ambiguous mechanisms of drug actions, time-consuming processes, high costs, and the risk of failure in drug discovery. These factors contribute to the lack of available protectors both for space radiation and RT. Further elucidation of the mechanisms by which PTMs participate in DSB repair and the identification of potential drugs against IR are urgently needed. In the future, advances in taking full use of multi-omics, theoretical prediction, and drug repositioning will offer a better risk-versus-reward trade-off. Although the current research on radioprotectors is insufficient and unclarified, systematic analyses of PTMs and the prioritization of strategies in drug R&D will show their potent power in target and drug discovery against cosmic radiation and facilitate improvements in radioprotection technology.

## Figures and Tables

**Figure 1 ijms-24-07656-f001:**
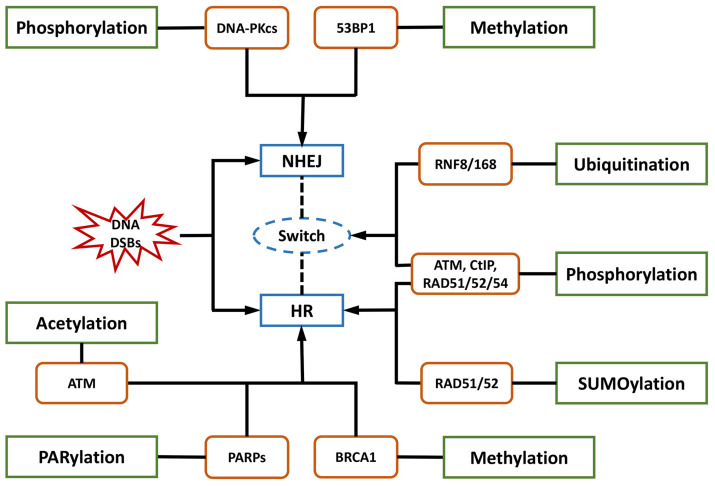
Post-translational modifications (PTMs) and their representative essential factors in regulation of non-homologous end joining (NHEJ) and homologous recombination (HR) in response to ionizing radiation (IR)-induced DNA damages. DSB = double-strand break.

**Figure 2 ijms-24-07656-f002:**
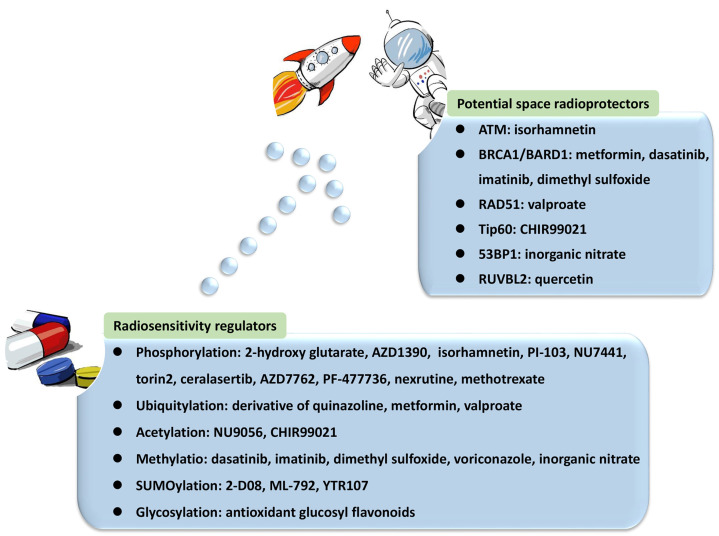
Compounds targeting post-translational modifications (PTMs) in DNA damage response (DDR) as radiosensitivity regulators, from which potential space radioprotectors may emerge.

**Table 1 ijms-24-07656-t001:** Essential factors in post-translational modifications (PTMs) in repair of DNA double-strand breaks (DSBs), their cellular functions, and application in radiosensitization and radioprotection.

Post-Translational Modification (PTM)	Essential Factor	Description	Application	Reference
Phosphorylation	ATM	⬤ Participates in phosphorylation of other proteins	⬤ Radiosensitization ⬤ Radioprotection	[86,87,88,89,90,91,92,93]
	DNA-PKcs	⬤ Plays roles in DSB repair and development of radioresistance	⬤ Radiosensitization	[88,94,95,96]
	CtIP	⬤ Controls the balance between NHEJ and HR ⬤ Controls DNA end resection	⬤ Radiosensitization	[42,97,98,99]
	ATR	⬤ Phosphorylates BRCA1 fragments ⬤ Implicated in DDR	⬤ Anticancer ⬤ Radiosensitization	[100,101]
	CHK1	⬤ Interacts with ATR ⬤ Implicated in DDR	⬤ Radiosensitization	[101,102,103]
	RAD51/52/54	⬤ Mediates DNA damage signal transduction ⬤ Central for HR	⬤ Anticancer ⬤ Radiosensitization	[43,70,104,105,106,107]
	H2AX	⬤ Regulates checkpoint pathways ⬤ Facilitates 53BP1 recruitment	/	[108,109]
Ubiquitylation	RNF8	⬤ Mediates UBC13-dependent ubiquitylation ⬤ Interacts with MDC1	⬤ Anticancer	[110,111,112]
	RNF168	⬤ Catalyzes histone ubiquitination ⬤ Assembles 53BP1 and BRCA1	⬤ Radiosensitization	[47,110,111,113,114,115,116]
	REV1	⬤ Modulates translesion synthesis ⬤ Involved in DNA damage tolerance	⬤ Radiosensitization	[117,118,119,120]
	MDM2	⬤ Affects p53 polyubiquitination ⬤ Keeps E3 ligase activity	⬤ Radiosensitization ⬤ Anticancer	[89,121,122,123,124]
	BRCA1/BARD1	⬤ Stimulates HR ⬤ Implicated in formation of cancers	⬤ Radioprotection	[83,125,126,127]
	RAD51	⬤ Radio-bidirectional effect	⬤ Radiosensitization ⬤ Radioprotection	[128]
Acetylation	Tip60	⬤ Involved in activation of ATM ⬤ Interacts with histone H3	⬤ Radiosensitization ⬤ Radioprotection	[90,129,130,131]
Methylation	BRCA1	⬤ Plays regulatory roles in radiosensitization and radioprotection	⬤ Radiosensitization ⬤ Radioprotection	[132,133,134,135]
	53BP1	⬤ Associated with ROS	⬤ Radiosensitization ⬤ Radioprotection	[136,137]
PARylation	PARP1	⬤ Recruited to DSB sites ⬤ Important in feedback regulation of PARylation	⬤ Anticancer	[138,139]
Neddylation	NEDD8	⬤ Recruitment and conjugation are highly dynamic	⬤ Anticancer	[140,141,142,143]
SUMOylation	/	⬤ Inhibits SUMOylation	⬤ Radiosensitization	[144]
	MEIIL3	⬤ Involved in colorectal cancer	⬤ Anticancer	[145]
	NPM1	⬤ Interacts with RAD51 and impairs DSB repair	⬤ Radiosensitization	[146]
Glycosylation	OGT	⬤ Regulates O-linked β-N-acetylglucosamine glycosylation	⬤ Anticancer	[147,148]
	NEIL3	⬤ Essential in DNA ICL repair	⬤ Radiosensitization	[149,150,151,152]
	RUVBL1/2	⬤ Indispensable for NEIL3 pathway	⬤ Radioprotection	[150,153]
Kcr	CDYL1	⬤ Edits histone code	/	[154]
	CBP/P300	⬤ Histone crotonyltransferase	⬤ Radiosensitization	[155,156,157,158]

DSB = double-strand break, NHEJ = non-homologous end joining, HR = homologous recombination, DDR = DNA damage response, ROS = reactive oxygen species, PARylation = poly ADP-ribosylation, OGT = O-GlcNAc transferase, ICL = interstrand cross-link, Kcr = lysine crotonylation.

**Table 2 ijms-24-07656-t002:** The current situation and future directions of development of radioprotectors for human space exploration.

Era	Condition and Tendency
Current	⬤ Insufficiency of available radioprotectors ⬤ Mechanisms of DDR are unclarified ⬤ Difficulties in traditional paradigm of drug discovery
Future	⬤ Progression in multi-omics ⬤ Rational computing analyses and designs ⬤ Strategy of drug repositioning ⬤ Combinations of drugs or targets

DDR = DNA damage response.

## Data Availability

Not applicable.

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
