# Peer review of "Drug Discovery Targeting Post-Translational Modifications in Response to DNA Damages Induced by Space Radiation"

_ijms, 2023, doi:10.3390/ijms24087656_

Round 1

Reviewer 1 Report

Review

IJMS-2276565

“Drug Discovery Targeting Post-Translational Modifications in Response to DNA Damages Induced by Space Radiation”

Dafei Xie, Qi Huang and Pingkun Zhou

Dear Editor, Dear Authors,

In the following review article, the authors have generated an attempt to summarize the literature and emphasize the importance of protein post-translational modifications (PTMs) in the control and efficiency of DNA repair pathways involved in the elimination of double-strand break (DSB), induced by space radiation. Keywords such as “DNA double-strand break”, “post-translational modification” and “space radiation” have been utilized by the authors in order to identify potential data entries. In the following review, authors have focused on PTM  (phosphorylation, ubiquitylation, acetylation, methylation and PARylation)  generated during DNA damage response (DDR). Most of the PTMs occurred in the important ionizing radiation targets (ATM, DNA-PKcs, CtIP, MDM2, and multiple ubiquitin ligases) have been described in detail. According to the authors, the following review with provide a repository of DDR candidate targets for therapeutic intervention. Moreover, the authors proposed new perspectives for the development of potential radioprotectors mitigating the effect of space radiation, thus allowing the extension of the duration of space travel.

Despite that the review article includes a significant amount of information and indeed could serve as a great source, providing rationale explaining the space radiation effects on genome integrity, there are still multiple points, which required further attention and need to be addressed.

1.    The major problem of the following review article is the lack of conceptual integrity. Most of the chapters are written by including sentences from the selected articles, which do not follow any structure. For instance, at many locations authors provide facts about phosphorylation, then for ubiquitination afterward again come back to phosphorylation and there is no unified template, which revealed the significance of PMTs in an ordered manner. There is a very helpful table, which depicts and summarizes the literature findings for specific PTMs processes and includes selected DDR targets. It is highly recommended that authors follow even the included table and express the provided information in a more organized manner.

2.     Second major cons of the review is the grammar and language style. Many sentences and even entire paragraphs are very difficult to understand as the grammar and the expressions are not entirely correct. This severely compromises the clarity of the review and limits the interest of the potential audience. It is highly recommended the review undergo extensive language correction and style improvements before the paper is considered for publication in the journal.

3.    Despite that authors have identified so many entries in the selected library, there are still multiple canonical PMTs, which are not included in the article. It will make the review more complete and attractive if such data is also mentioned.

4. Figure 1 is conceptually not accurate and should be edited. Figure report that the activation of DSB repair pathways controls the PMTs, but in many cases, it is also true that the PMTs are in fact the controlling mechanisms of DSB repair mechanisms. Moreover, there is no indication of PMTs involvement in the regulation of alt-EJ and SSA. Also, no indication of PTMs, playing a role in the Rad51 and Ku70/80 axes of DSB repair.

5. Figure 2 is obsolete, and does not provide useful information. It could be converted into a table. However, some other key mechanisms explained by the authors could be translated into figures.

Reviewer 2 Report

Xie and colleagues provide a rather comprehensive literature review about current state of the scientific knowledge and research in regard to drug discovery targeting post-translational modification in response to DNA damage induced by cosmic ionizing radiation. Owing growing pursuit of human space exploration beyond the lower Earth orbit and the Moon, the topic of the review is of an emerging attention to aid future human radioprotection strategies in deep space. Generally, the Authors are to be applauded for efforts to position and discuss subject of the review in a broad context of available literature data. Minor points to improve the manuscript encompass enhancing the clarity of the narrative and perceptibility of the Table 1 (too crowded and expanding beyond the margins of the layout). 

Round 2

Reviewer 1 Report

Dear Editor, Dear Authors, 

In the current version of the manuscript, all previous concerns and issues have been addressed by the authors, which renders the review article suitable for publication in the journal. Therefore, I am prone to suggest the submitted review article "Drug Discovery Targeting Post-Translational Modifications in Response to DNA Damages Induced by Space Radiation" by Xie et al. for publication in the journal.